# An Inflammatory Signature to Predict the Clinical Benefit of First-Line Cetuximab Plus Platinum-Based Chemotherapy in Recurrent/Metastatic Head and Neck Cancer

**DOI:** 10.3390/cells11193176

**Published:** 2022-10-10

**Authors:** Stefano Cavalieri, Mara Serena Serafini, Andrea Carenzo, Silvana Canevari, Deborah Lenoci, Federico Pistore, Rosalba Miceli, Stefania Vecchio, Daris Ferrari, Cecilia Moro, Andrea Sponghini, Alessia Caldara, Maria Cossu Rocca, Simona Secondino, Gabriella Moretti, Nerina Denaro, Francesco Caponigro, Emanuela Vaccher, Gaetana Rinaldi, Francesco Ferraù, Paolo Bossi, Lisa Licitra, Loris De Cecco

**Affiliations:** 1Head and Neck Medical Oncology Department, Fondazione IRCCS (Istituto di Ricovero e Cura a Carattere Scientifico) Istituto Nazionale dei Tumori, 20133 Milan, Italy; 2Department of Oncology and Hemato-Oncology, University of Milan, 20122 Milan, Italy; 3Molecular Mechanisms Unit, Department of Research, Fondazione IRCCS Istituto Nazionale dei Tumori, 20133 Milan, Italy; 4Fondazione IRCCS Istituto Nazionale dei Tumori, 20133 Milan, Italy; 5Clinical Epidemiology and Trial Organization, Fondazione IRCCS Istituto Nazionale dei Tumori, 20133 Milan, Italy; 6Medical Oncology, IRCCS Ospedale Policlinico San Martino, 16132 Genova, Italy; 7Medical Oncology, Ospedali Santi Paolo e Carlo, 20142 Milan, Italy; 8Medical Oncology, Azienda Ospedaliera Papa Giovanni XXIII, 24127 Bergamo, Italy; 9Medical Oncology, A.O. Universitaria Maggiore della Carità, 28100 Novara, Italy; 10Medical Oncology, Ospedale Santa Chiara, 38122 Trento, Italy; 11Division of Urogenital and Head and Neck Medical Oncology, European Institute of Oncology IRCCS, 20133 Milan, Italy; 12Medical Oncology, Fondazione IRCCS Policlinico San Matteo, 27100 Pavia, Italy; 13Azienda USL-IRCCS di Reggio Emilia, 42122 Reggio Emilia, Italy; 14Medical Oncology, St. Croce e Carle University Teaching Hospital and ARCO Foundation, 12045 Cuneo, Italy; 15Medical Oncology, Istituto Nazionale Tumori—IRCCS—Fondazione Pascale, 80131 Naples, Italy; 16Medical Oncology and Immune-Related Tumours, Centro di Riferimento Oncologico di Aviano (CRO) IRCCS, 33081 Aviano, Italy; 17Medical Oncology, AOU Policlinico “Paolo Giaccone”, 90127 Palermo, Italy; 18Medical Oncology, Ospedale San Vincenzo, 98039 Taormina, Italy

**Keywords:** HNSCC, targeted therapy, cetuximab, gene-expression, immune biomarkers, inflammatory signature

## Abstract

Epidermal growth factor receptor (EGFR) pathway has been shown to play a crucial role in several inflammatory conditions and host immune-inflammation status is related to tumor prognosis. This study aims to evaluate the prognostic significance of a four-gene inflammatory signature in recurrent/metastatic (R/M) head and neck squamous cell carcinoma (HNSCC) patients treated with the EGFR inhibitor cetuximab plus chemotherapy. The inflammatory signature was assessed on 123 R/M HNSCC patients, enrolled in the multicenter trial B490 receiving first-line cetuximab plus platinum-based chemotherapy. The primary endpoint of the study was progression free survival (PFS), while secondary endpoints were overall survival (OS) and objective response rate (ORR). The patient population was subdivided into 3 groups according to the signature score groups. The four-genes-signature proved a significant prognostic value, resulting in a median PFS of 9.2 months in patients with high vs. 6.2 months for intermediate vs. 3.9 months for low values (*p* = 0.0016). The same findings were confirmed for OS, with median time of 18.4, 13.4, and 7.5 months for high, intermediate, and low values of the score, respectively (*p* = 0.0001). When ORR was considered, the signature was significantly higher in responders than in non-responders (*p* = 0.0092), reaching an area under the curve (AUC) of 0.65 (95% CI: 0.55–0.75). Our findings highlight the role of inflammation in the response to cetuximab and chemotherapy in R/M-HNSCC and may have translational implications for improving treatment selection.

## 1. Introduction

Recurrent head and neck squamous cell carcinoma (HNSCC) not amenable to loco-regional salvage therapies and metastatic HNSCC are treated with systemic approaches [1]. The treatment choice relies on patient- and disease-related factors, including patient’s performance status, comorbidities, disease-free interval, and previous therapies.

After the publication of the Keynote048 trial results [2], the state-of-the-art option for the so-called platinum-sensitive recurrent/metastatic (R/M) has changed. To date, according to international guidelines [1], patients with a combined positive score (CPS) ≥ 1 may receive pembrolizumab monotherapy or platinum-based chemotherapy and 5-fluorouracil (5-FU) plus immunotherapy with pembrolizumab. In contrast, subjects with a PD-L1-negative disease (CPS = 0), accounting for 15% of the R/M HNSCC patient population [2], are treated with platinum-based chemotherapy plus cetuximab. Among the available therapeutic options for PD-L1-negative R/M HNSCC, one of the available regimens is Extreme, which is made of the combination of cetuximab, cisplatin (or carboplatin for cisplatin-unfit subjects) and 5-fluorouracil (5-FU). In alternative regimens, 5-FU may be replaced by taxanes, either docetaxel or paclitaxel [1], or it might be omitted [3].

Monoclonal antibody-guided targeted therapy, such as anti-EGFR, and immune checkpoint inhibitors (ICI), alone or in combination with chemotherapy, have been approved for R/M HNSCC by FDA/EMA but, at variance to other types of tumors (i.e., colon cancer for cetuximab or lung and melanoma for ICIs), the percent of long-term responding patients is lower. To date, biomarkers to identify patients that could benefit from the therapy (predictive of therapy response and/or better overall survival) are still under investigation. Previously, we have shown that patients achieving long-lasting responses to the combination of chemotherapy and cetuximab showed a profile enriched in strong EGFR signaling phenotype and hypoxic differentiation [4].

Moreover, it is well-known that cetuximab plays a role in modulating the immune system in cancer [5,6,7,8,9,10]. As an example, proof of concept studies identified that inflammatory biomarkers can predict treatment response and favorable survival in patients who underwent first-line cetuximab plus chemotherapy in metastatic colorectal cancer [11,12]. However, while inflamed tumor microenvironment gene expression signatures are under evaluation as predictive biomarkers to ICI response in other type of cancers [13,14], at present, in the field of HNSCC, the relationship between inflammatory biomarkers, early treatment response, and cetuximab efficacy still needs to be elucidated.

With the present analysis, we aimed to explore the prognostic role of an inflammation signature in an R/M HNSCC patient population treated with a first-line cetuximab-based therapy in a multicenter phase II clinical trial [3].

## 2. Materials and Methods

### 2.1. Study Design and Case Material

Gene expression analysis was conducted on formalin-fixed paraffin-embedded (FFPE) primary tumor specimens of patients included in the multicenter B490 phase II randomized trial (Clinical trial number EudraCT# 2011-002564-24), in which patients received cetuximab plus cisplatin with/without paclitaxel (hereafter CetCis versus CetCisPac). For the present analysis, patients were selected for the availability of a primary tumor specimen, clinical information on response, and follow-up data. Detailed methods about eligibility criteria and treatments were described in the main paper of the clinical study [3]. The primary endpoint of the study was progression-free survival (PFS), defined as the time between the date of randomization and the date of progression or death without evidence of progression, whichever occurred first. Secondary endpoints were: (i) overall survival (OS), defined as the interval between the date of randomization and that of death from any cause; (ii) objective response rate (ORR), according to Response Evaluation Criteria in Solid Tumors (RECIST, Version 1.1). Follow-up data were updated in July 2020. This phase II study was designed as a non-inferiority study between CetCis and CetCisPac, with PFS as a primary endpoint. The non-inferiority was met and the final results showed no statistically significant differences in terms of PFS, OS and best objective response rate. For these reasons, we aggregated the data of the two arms for the analysis of PFS, OS, and the best objective response. The study sponsor’s ethical committee approved the conduction of the clinical trial on the 29 July 2011 (local study identifier INT35-11) and the translational analyses on the 17 December 2013 (PG/U 0013329). All patients provided written informed consent for translational research prior to study entry.

### 2.2. Gene Expression Analysis

The case material is based on primary tumors only collected at the first diagnosis and archived as FFPE blocks. Since the specimens were retrieved for the translational research purposes of the present study at the time of patient’s accrual in the trial, their storage was in the range of 1–3 years. Being aware that FFPE processing and tissue storage could result in highly degraded RNAs which might impair gene expression-based biomarker discovery by RNAseq, we applied pre- and post-analytical quality checks to address these issues [15]. After histopathological revision by an expert pathologist, selected tumor areas from FFPE blocks were manually macro-dissected and total RNA was immediately extracted. RNA extraction was performed using the Qiagen RNeasy Mini Kit with the QIAcube robotic station (Qiagen, Düsseldorf, Germany), according to the manufacturer’s recommendations. Quantification and quality check were performed using the Qubit 3.0 fluorometer (Life Technologies, Carlsbad, CA, USA) and the TapeStation 4200 system (Agilent Technologies, Santa Clara, CA, USA). Transcriptome libraries were generated from 100 ngr of total RNA with the TruSeq RNA Access Library Prep Kit (Illumina, San Diego CA, USA) following the manufacturer’s instructions. Libraries were pooled (8 samples/pool), denatured, clustered onto a high output sequencing flow cell (V2 chemistry), and sequenced on NextSeq500 (Illumina, San Diego CA, USA) in paired-end mode with a read length of 2 × 75 bp generating 50 million reads/sample.

### 2.3. Bioinformatic Analysis

Raw reads were aligned to the human reference genome assembly GRCh38 through the align () function of the Rsubread R package, version 2.0.1. A gapped index was built for the reference genome through the Rsubread function buildindex. BAM files were successively used as input to the featureCount function in order to obtain a raw count matrix. Then, the variance stabilizing transformation (vst), in the DESeq2 R package was applied to normalize the raw counts.

We assessed the value of a four-gene inflammatory signature, previously developed based on literature data [16,17] and patented by Bristol Myers Squibb (Princeton, NJ, USA) (patent numbers WO2020/198672 and WO2020/198676). The expression values of the four-gene inflammatory signature, which included CD274, CD8A, LAG3, and STAT1, were retrieved from the normalized data matrix. Following the signature data processing, the gene expression values were scaled and combined to obtain a score. The signature does not include any weight for the 4 genes that were simply scaled and combined; thus, each gene contributed equally to the signature. The four-gene signature generates a continuous score that is associated with the inflammation level assessed by the four genes (i.e., high score -> high inflammation); the scores were stratified based on tertiles. Further details about this inflammatory signature and its application to other types of cancer are reported elsewhere [13,14].

### 2.4. Statistical Analysis

According to the groups of the gene signature scores, patients were stratified into three groups: high, intermediate, and low score. We estimated the signature stratification capability by the Kaplan–Meier method and compared the PFS and OS curves with log-rank test. Survival analysis and visualization were provided using survminer R package and ggsurvplot function. Uni- and multivariable Cox regression analyses were used to test the effect of the signature on PFS and OS by adjusting for ECOG (Eastern Cooperative Oncology Group) performance status (PS), an objective clinical scale to evaluating the overall clinical conditions of cancer patients assessing their activities of daily life and primary HNSCC site (oropharynx versus others) by using survminer and survival R packages. Results of the Cox analyses are reported in terms of hazard ratios (HR), corresponding 95% confidence intervals (95% CI) along with *p*-values at two-sided Wald test. ORR (CR or PR) was assessed using descriptive statistics, and the signature ability for discriminating responders from non-responders was measured by estimating the area under the receiver operating characteristic curve (ROC-AUC) using plotROC R package. All the statistical analyses were performed using R version 3.6.0. In all cases, statistical significance was set at 0.05.

## 3. Results

### 3.1. Patient Characteristics

A total of 201 patients were enrolled and 10 cases were excluded from the intention-to-treat analysis [3]. Out of the 191 randomized patients (100 CetCis arm; 91 CetCisPac arm), FFPE specimens of primary tumor were available and suitable for RNA extraction in 123 cases (64%), which were all included in survival analysis (PFS and OS) (Figure 1). Characteristics of patients with available RNA were consistent with the whole study population (Table 1) [3].

### 3.2. Clinical Outcomes

After the survival data update in 2020, the median follow-up in the study cohort (123 patients) was 52.9 months (95% CI 26.7–74.9), and median PFS was 6.1 months (95% CI 4.9–7.2). PFS at landmark times 3, 6, 9, and 18 months were 76%, 51%, 32%, and 14%, respectively. Median OS was 12.4 months (95% CI 9.4–14.1). OS at 9, 12, and 18 months were 64%, 53%, and 36%, respectively. ORR was 45% and 61% in the CetCis and the CetCisPac arms, respectively (OR = 0.52; 95% CI: 0.24–1.1, *p* = 0.086 by Cochran–Mantel–Haenszel test). Apart from a longer median follow-up, these findings were consistent with the ones previously reported in the paper published in 2017 [3], and both primary and secondary endpoints confirm no difference between the two regimens.

### 3.3. Four-Gene Inflammatory Signature

Subdividing the patient population into three groups, according to the expression of the four-gene inflammatory signature, PFS was longer in patients with a higher inflammatory score than in those with lower score: median PFS 9.2 months (95% CI: 7.6–14.8) vs. 6.2 months (95% CI 5–8) vs. 3.9 months (95% CI: 3.2–5.5) in high, intermediate, and low score, respectively (*p* = 0.0016; Figure 2A). In the same three groups, median OS was 18.4 months (95% CI: 12.6–27.5) vs. 13.4 months (95% CI: 10.3–25.8) vs. 7.5 months (95% CI: 5.5–10.4), *p* = 0.0001 (Figure 2B).

The univariable Cox models including the four-gene inflammatory signature as a continuous variable (Table 2) showed that high inflammation is associated with improved PFS and OS; in addition, multivariable Cox models proved that the inflammatory score is a significant prognostic factor for both PFS and OS independently of ECOG PS and primary HNSCC site. When the signature is stratified in high, intermediate, and high scores (Appendix A), the univariable analyses by Cox models showed that patients belonging to the high score group and with the highest signature expression experienced an improved PFS and OS compared with those in intermediate and low score groups and these results were confirmed at multivariable analyses by adjusting for ECOG PS (prognostic for PFS and OS) and primary HNSCC site (prognostic for OS only (Appendix A). This proves the prognostic value of the four-gene inflammatory signature in patients treated with cetuximab plus cisplatin-based chemotherapy.

ORR data were available in 112 cases (91%). The analyses were performed by comparing responders (subjects achieving either a complete or partial response, CR and PR respectively; n = 59) versus non-responders (subjects having either stable disease or disease progression as best response, SD and PD, respectively; n = 53). As a continuous variable, the studied inflammatory signature exhibited significantly higher values in responders than in non-responders (*p* = 0.0092; Figure 3A). The discriminative ability of the four-gene inflammatory signature reached a ROC AUC of 0.65 (95% CI: 0.55–0.75; Figure 3B).

## 4. Discussion

An inflamed phenotype seems important in predicting the response to cetuximab, whose activity is not limited to EGFR inhibition. Expression level and the copy number of EGFR failed to predict cetuximab response [18,19]. Cetuximab activity is known to be linked to its immune-modulating role, mediated mainly by antibody-dependent cellular cytotoxicity (ADCC) [5]. In the present work, we demonstrated that a four-gene inflammatory signature has an independent prognostic role in forecasting PFS and OS in R/M HNSCC patients first-line treated with cetuximab plus platinum-based chemotherapy. This prognostic role was independent of clinical factors, such as PS and primary HNSCC site. Moreover, the gene signature was associated with an objective response.

Our findings are consistent with what was observed in other malignancies treated with ICI. The investigated four-gene inflammatory signature reflects key functions in immune modulation, such as interferon-*γ*/STAT1-dependent CD^8+^ T-cell expansion (STAT1 and CD8A genes), LAG-3–dependent T-cell exhaustion (LAG3 gene), and high PD-L1 (CD274 gene) expression. Previously, the signature was found to be associated with ORR and OS in a cohort of hepatocellular carcinoma patients treated with nivolumab [13]. An analogous prediction of the benefit from ICI was observed in advanced melanoma patients treated with nivolumab, ipilimumab, or their combination [20,21]. Furthermore, the signature was shown to be predictive of response in gastric/gastroesophageal junction cancer patients treated with nivolumab +/− ipilimumab [14].

Recent studies have shown that the PD-L1 expression assessed by immunohistochemistry is positively correlated with its gene expression [20,21], CD274 copy number changes [22], and amplification [23]. Therefore, for our analysis, we assumed that the CD274 component of the proposed gene expression signature could be used as a surrogate for PD-L1 expression. In the HNSCC setting, from one side, several studies showed that PD-L1 is a strong prognosticator, and that its expression is higher in radiosensitive tumors [24]. On the other hand, a meta-analysis revealed that PD-L1 expression detected by immunohistochemistry was not prognostic for HNSCC but turned out to predict PFS in R/M only [25]. In the Keynote048 clinical trial [2], PDL1 expression was not a prognostic or a predictive factor in patients enrolled in the control arms treated with the Extreme regimen (cisplatin + 5-fluorouracil + cetuximab); thus, the prognostic role of PD-L1 in HNSCC is still under debate.

The involvement of CD8A gene in the proposed signature seems to confirm the well-known positive prognostic role of CD^8+^ tumor-infiltrating lymphocytes (TILs) in several cancers, including HNSCC [26]. In RAS wild-type colorectal cancer patients, cetuximab was shown to increase TILs together with PD-L1 and LAG3 upregulation [27]. We may hypothesize a similar mechanism in HNSCC as well, since RAS is very rarely mutated in this setting [28].

Another component of the studied gene signature is STAT1. In this context, the immune escape mediated by STAT1 inhibition and STAT3 activation can be a downstream pathway of EGFR [29]. Moreover, de-inhibition of STAT1 has been shown to be involved in HNSCC immune evasion. This hypothesis is supported by the fact that JAK2/STAT pathway, together with its interplay with IFN-γ, plays a strong role in both EGFR-mediated immune escape and PD-L1 upregulation [30]. Moreover, in preclinical models, the inhibition of EGFR has been shown to trigger STAT1 activation, thus enhancing the adaptive cellular immunity [31]. Therefore, the hypothesis that cetuximab activity could be related to immune-modulating activity is supported further.

In the R/M HNSCC setting, the clinical activity and the prolonged responses observed with the association of cetuximab and the anti-PD1 pembrolizumab seems to corroborate the validity of this hypothesis [32]. However, we have to admit a limitation in the current study which relied on the high percent of patients (around 40%) with no sufficient amount of tissue or poor quality of extracted RNA that precluded deeper molecular analyses. Since CPS is equal or higher than 1 in approximately 85% of R/M HNSCC patients [2], as per standard-of-care, the majority of subjects were treated in first line with pembrolizumab alone or chemo-immunotherapy [1]. However, the remaining 15% of cases were still treated with platinum-based chemotherapy plus cetuximab. For these patients, the results of this study will create a future direction and the opportunity to better decipher the biology of the disease, and the benefit of the systemic treatment. Our findings warrant the design of a dedicated prospective trial. In this setting, other gene expression signatures have demonstrated their role in predicting cetuximab activity in HNSCC, notably the Cl3-hypoxia group of six-cluster model proposed and validated by our group [4]. Differently from the four-gene inflammatory signature, this cluster is characterized by several onco-signatures, notably EGFR and RAS, and altered pathways, such as hypoxia over-expression, but not by immune system pathways. Therefore, it is possible that response to chemotherapy and cetuximab may be triggered by different pathways in HNSCC, with one component linked to EGFR trait and hypoxia and another one to inflammation and immune response. Further evaluations and a combination of the aforementioned signatures could potentially enable better forecasting of the response to cetuximab and immune-related treatments.

## 5. Conclusions

To our knowledge, the present work reported the first analysis of the four-gene inflammatory signature with respect to the clinical efficacy of cetuximab plus chemotherapy in HNSCC, suggesting its prognostic and predictive role in a newly analyzed clinical setting. We may anticipate that similar results might be observed in an HNSCC patient population treated with immunotherapy, alone or in combination with cetuximab. Further studies are needed to confirm this hypothesis.

## Figures and Tables

**Figure 1 cells-11-03176-f001:**
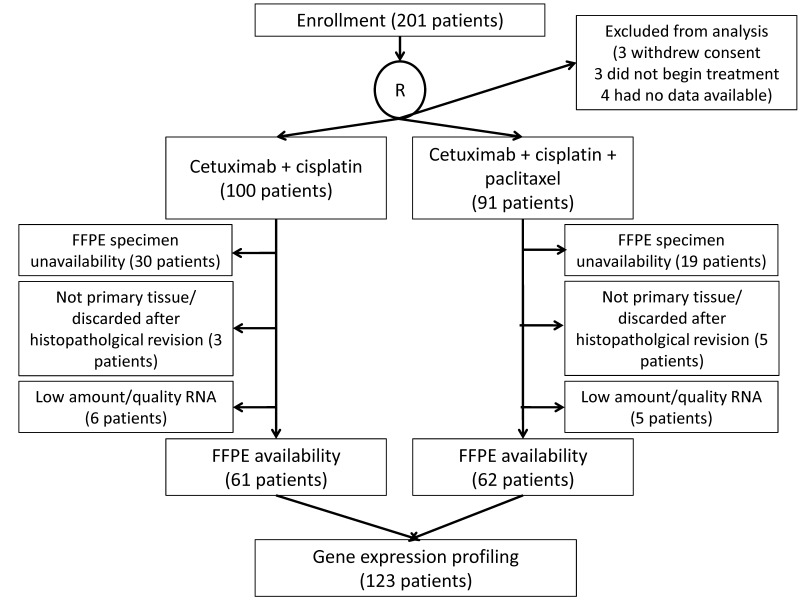
CONSORT diagram.

**Figure 2 cells-11-03176-f002:**
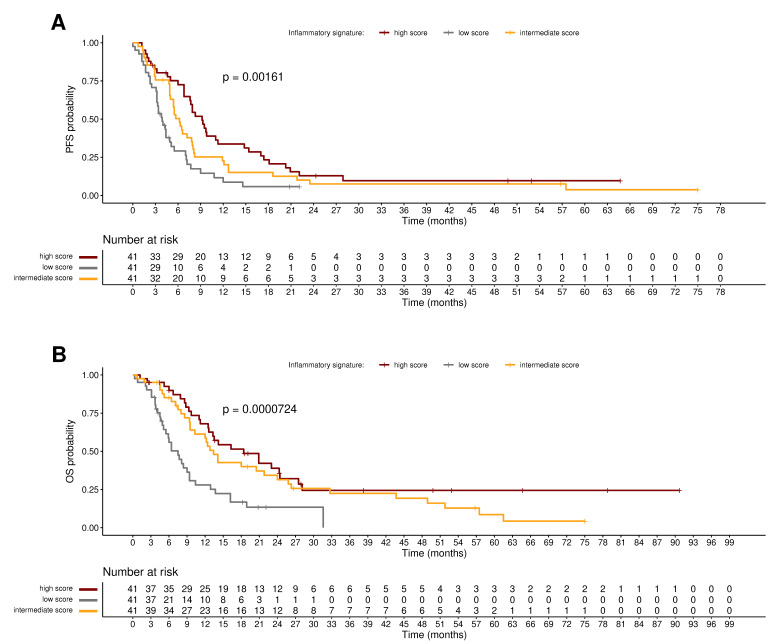
Prognostic value of the inflammation signature. (**A**) Kaplan–Meier curves of B490 patients’ PFS associated to the four-gene inflammatory signature. (**B**) Kaplan–Meier curves of B490 patients’ OS associated to the four-gene inflammatory signature. Curve separation was assessed by the log-rank test. Samples were stratified based on tertiles: high (red), intermediate (yellow) and low (gray) inflammatory score groups.

**Figure 3 cells-11-03176-f003:**
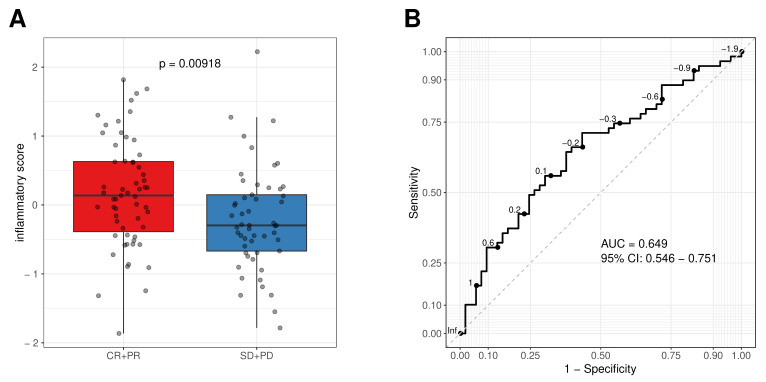
The inflammatory score as continuous variable was associated to cetuximab response. (**A**) Boxplot of the inflammatory scores in responders (complete responders + partial responders, CR + PR) and non-responder (stable disease + progressive disease, SD + PD). The inflammatory score is based on scaled expression of the four genes retrieved from normalized RNAseq data, (**B**) receiver operating characteristics (ROC) curve and area under the curve (AUC) of the four-gene inflammatory signature.

**Table 1 cells-11-03176-t001:** Clinical-pathological characteristics of B490 study (n = 191) compared to the cohort considered in the current study with available RNA (n = 123) that was analyzed for gene expression.

Variable	Study Population	Clinical Trial Article (n = 191)	Samples with Available RNA (n = 123)
Randomization arm	Cetuximab + cisplatin Cetuximab + cisplatin + paclitaxel	100 (52%) 91 (48%)	61 (50%) 62 (50%)
Age (years)	Median Range	63 33–83	62 33–83
Sex, n (%)	Female Male	42 (22%) 149 (78%)	27 (22%) 96 (78%)
ECOG PS, n (%)	0 1	97 (51%) 94 (49%)	74 (60%) 49 (40%)
Primary tumor site, n (%)	Oropharynx Other	70 (37%) 121 (63%)	49 (40%) 74 (60%)
HPV, n (%) (oropharynx only)	Not tested HPV-negative HPV-positive	41 (59%) 16 (23%) 13 (19%)	35 (71%) 6 (12%) 8 (16%)
Site of recurrence, n (%)	Missing Local recurrence Locoregional recurrence Metastasis Metastasis/local/regional recurrence Regional recurrence	3 (2%) 27 (14%) 42 (22%) 59 (31%) 49 (26%) 11 (6%)	1 (1%) 14 (11%) 26 (21%) 37 (30%) 38 (31%) 7 (6%)

**Table 2 cells-11-03176-t002:** Univariate and multivariate cox proportional hazard regression analysis of the four-gene inflammatory signature (continuous variable), considering overall survival and progression free survival.

		**Univariable Analysis**		**Multivariable Analysis**	
**Overall Survival (OS)**		**HR (95% CI)**	***p*-Value**	**HR (95% CI)**	***p*-Value**
**inflammatory signature**	continuous variable	0.66 (0.53–0.79)	0.001	0.65 (0.51–0.82)	<0.001
**ECOG PS**	1 vs. 0	1.28 (1.08–1.48)	0.213	1.51 (1.02–2.25)	0.041
**subsite**	oropharynx vs. other	1.52 (1.33–1.72)	0.031	1.58 (1.07–2.34)	0.022
		**Univariable analysis**		**Multivariable analysis**	
**Progression Free Survival (PFS)**		**HR (95% CI)**	***p*-value**	**HR (95% CI)**	***p*-value**
**inflammatory signature**	continuous variable	0.57 (0.43–0.72)	<0.001	0.54 (0.4–0.71)	<0.001
**ECOG PS**	1 vs. 0	1.39 (1.18–1.6)	0.117	1.80 (1.2–2.77)	0.008
**subsite**	oropharynx vs. other	1.31 (1.1–1.52)	0.193	1.36 (0.9–2.07)	0.146

## Data Availability

The authors declare that the data are available upon reasonable request.

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
