# Peer review of "An Inflammatory Signature to Predict the Clinical Benefit of First-Line Cetuximab Plus Platinum-Based Chemotherapy in Recurrent/Metastatic Head and Neck Cancer"

_cells, 2022, doi:10.3390/cells11193176_

Round 1
Reviewer 1 Report
This study is an interesting and valuable contribution to the existing literature, and indeed there is a need for predictive biomarkers for response to (targeted) therapy. I do have some comments, mainly asking the authors to be more precise or methods and terminology.
Comments
1. In the title and throughout the text, the authors speak of '...first-line cetuximab plus platinum-based chemotherapy in recurrent/metastatic head and neck cancer...' but if I understand well, the patient series concern primary HNSCC, which have been followed and PFS (development of recurrence or metastasis) and OS and ORR have been analyzed. Also, the 4-gene expression signature was analyzed in specimens from primary tumors. Please check this carefully.
2. Please also explain more clearly, already in Introduction (but shorter than you did in Discussion), how you arrived at these 4 genes. Was it based on similar 'immuno' signatures in the literature? Similar in the sense of gene expression studies of these individual genes, or perhaps protein expression studies?
3. As you have a patent on this gene expression panel, maybe it's good to discuss more details of it? Were all 4 genes equally important? Will the panel also be useful for other tumors than HNSCC?
4. In methods, please explain and justify how high, intermediate and low scores of the inflammation signature were defined. According to figure 2 it is a continuous variable? Also, the y-axis of figure 2A has no units
5. Table 1 specifies 61 patients receiving Cetuximab+cisplatin and 62 Cetuximab+cisplatin+paclitaxel; have the authors studied PFS, OS and ORR separately for these two groups? Were their responses different? It may useful to justify why you analysed all 123 patients together.
6. Similarly, have you studied PFS, OS and ORR separately for oropharynx and other-site tumors? As it is known that generally orophraynx SCC have better outcomes, it may be useful to address this topic, perhaps add an analysis in Suppl data.
7. In the first line of abstract and in the text you state that the immuno-modulating role of cetuximab 'has been shown' or 'is well-known'. Nonetheless you only give two references (5,6), which seem few...
8. Third and fourth paragraph of discussion speaks of the prognostic role of PD-L1 and CD8+ TILs (references 17-18-19). Could something more be said if this is so with the same treatment schemes as the study of this present manuscript (Cetuximab+cisplatin and 62 Cetuximab+cisplatin+paclitaxel)?
9. Fifth paragraph of discussion states: 'By observing the results of the current study, we cannot exclude 242 that cetuximab clinical activity might be owed to its indirect role on STAT1 de-inhibition...' Which exact results are referred to? Expression score of the 4 gene signature or the expression of only STAT1?
Minor points:
- Are Table 2 and Suppl table 1 the same?
- In abstract there are many abbreviations not explained.
- ECOG performance status may need explanation for non-clinicians? Also its abbreviation?
Reviewer 2 Report
Dear authors
I have read your manuscript concerning the prognostic inflammatory signature for patients R/M SCCHN receiving cetuximab + platin-based chemotherapy as first-line therapy.
Novel and robust biomarkers for therapy selection and prediction of response / outcome are an unmet need and I would like to congratulate the authors on the very interesting research article.
Herewith, I provide you with some additional comments in order to imorve the manuscript.
MATERIALS AND METHODS
1) Line 105: you indicate that FFPE of primary tumors was collected. Are this freshly taken specimens? Or also archival tissue from e.g. primary resection of the tumor? This because time between specimen collection and analysis / therapy can result in significant bias, as has already been proven for IHC of FFPE as for RNA extraction of FFPE tissue. I would therefore recommend to elaborate more on the time between specimens collection and time of analysis in the Materials and methods section, as well as include a comment in the Discussion section.
RESULTS
2) Table 1: you only give the HPV numbers for oropharyngeal carcinoma. I know that HPV is not so common in other SCCHN indications, but was HPV measured for these other indications or not? OF not, I would add these numbers to the row “missing”. If known, I would indicate the numbers for the entire cohort described in the manuscript.
3) Table 2: it is not clear what is the difference between 2A and 2B. I presume 2A is OS and 2B is PFS? Please clarify.
4) Table 2: why do you perform an analysis for the SCCHN subsite and why do you select oropharynx? Is this because of the higher prevalence of HPV infections? If so, this should be indicated and, preferentially, the analysis should be made for HPV positive versus HPV negative (e.g. performing additional stainings for all HPV stainings that are missing).
5) Line 195-198: you correctly use the inflammatory signature as a continuous variable. Did you attempt to calculate a cut-off value for the prognostic and predictive properties of the signature? This as categorical variable is more easily implemented into daily clinical practice.
DISCUSSION
6) Line 217 + 245: a spirale is indicated in the text following interferon. Please adapt. Furthermore, you use IFN-γ on line 245. Please be consistently.
7) Line 260-262: I missing some additioanl future perspectives as well as limitations of the current studies (if any).
Reviewer 3 Report
The ms. will be improved if authors address the followings:
1. Transfer supplemental Fig. 1 to the main ms. section.
2. Table 1: please correct paclitaxel (missing "l"). What does "Missing" refer to for HPV - Not detected?
3. It appears that the two treatment groups (+/- paclitaxel) were taken together for the 4-gene signature prediction. Are the results different if they are analyzed separately.
4. Is PD-L1 in the high score group a marker for increased EGFR activity?
